# Hyperspectral Image Classification with the Orthogonal Self-Attention ResNet and Two-Step Support Vector Machine

Heting Sun, Liguo Wang *, Haitao Liu and Yinbang Sun

College of Information and Communication Engineering, Dalian Minzu University, Dalian 116600, China;
sunheting@dlnu.edu.cn (H.S.); lhtdlnu@dlnu.edu.cn (H.L.); 20211105296@dlnu.edu.cn (Y.S.)
* Correspondence: wangliguo@hrbeu.edu.cn

**Abstract:** Hyperspectral image classification plays a crucial role in remote sensing image analysis by classifying pixels. However, the existing methods require more spatial–global information interaction and feature extraction capabilities. To overcome these challenges, this paper proposes a novel model for hyperspectral image classification using an orthogonal self-attention ResNet and a two-step support vector machine (OSANet-TSSVM). The OSANet-TSSVM model comprises two essential components: a deep feature extraction network and an improved support vector machine (SVM) classification module. The deep feature extraction network incorporates an orthogonal self-attention module (OSM) and a channel attention module (CAM) to enhance the spatial–spectral feature extraction. The OSM focuses on computing 2D self-attention weights for the orthogonal dimensions of an image, resulting in a reduced number of parameters while capturing comprehensive global contextual information. In contrast, the CAM independently learns attention weights along the channel dimension. The CAM autonomously learns attention weights along the channel dimension, enabling the deep network to emphasise crucial channel information and enhance the spectral feature extraction capability. In addition to the feature extraction network, the OSANet-TSSVM model leverages an improved SVM classification module known as the two-step support vector machine (TSSVM) model. This module preserves the discriminative outcomes of the first-level SVM subclassifier and remaps them as new features for the TSSVM training. By integrating the results of the two classifiers, the deficiencies of the individual classifiers were effectively compensated, resulting in significantly enhanced classification accuracy. The performance of the proposed OSANet-TSSVM model was thoroughly evaluated using public datasets. The experimental results demonstrated that the model performed well in both subjective and objective evaluation metrics. The superiority of this model highlights its potential for advancing hyperspectral image classification in remote sensing applications.

**Keywords:** hyperspectral image classification; orthogonal self-attention module; channel attention module; two-step support vector machine

## 1. Introduction

Compared with traditional RGB images, hyperspectral images offer nanometre-level spectral resolution and the ability to capture information from hundreds of bands. This provides them with a more robust feature recognition capability and higher classification reliability. Hyperspectral images contain rich spatial and spectral radiation information, which makes them comprehensive carriers of multiple data types. Quantified continuous spectral curve data lends itself favourably to image classification, making hyperspectral image classification a core technology in remote sensing applications. This task assigns unique category labels to each pixel based on the spectral information. Accurate classification results provide reliable data support for subsequent applications and decision-making. Over the past few decades, numerous hyperspectral image classification algorithms have

been proposed and have been widely applied to land monitoring [1,2], vegetation identification [3], and lake water quality detection [4].

Hyperspectral image classification algorithms can be categorised into two types: traditional and deep-learning-based methods. Conventional methods include the k-nearest neighbour classification (KNN) [5], decision trees [6,7], and support vector machines (SVM) [8,9]. Among these methods, SVM with kernel techniques has been widely studied and applied to classify high-dimensional and strongly correlated spectral data accurately. Cui et al. [10] proposed a recursive edge-preserving filtering algorithm as the preprocessing step. This algorithm combines bilateral and recursive filtering to remove noise while preserving the edge details. Liu et al. [11] introduced a nonparallel support vector machine algorithm with a constrained bias term and additional empirical risk minimisation to improve the generalisation ability of the model for unknown data. Mahendra et al. [12] proposed an efficient hardware architecture for the SVM classification of hyperspectral remote sensing data using high-level synthesis (HLS), enabling real-time classification systems for remote sensing applications. Although traditional SVM algorithms have achieved good classification performance and robustness for data with few samples, high dimensionality, and clear class boundaries, they are sensitive to noise and outliers, are prone to overfitting, and require manual parameter tuning, resulting in poor generalisation ability.

Deep neural networks (DNN) have remarkable capabilities in terms of non-linear expressions and autonomous learning parameters. These capabilities enable the extraction of feature information from large-scale datasets, making them widely applicable to computer vision. When it comes to hyperspectral image classification tasks, the mainstream deep backbone networks include convolutional neural networks (CNN) [13], recurrent neural networks (RNN) [14], generative adversarial networks (GAN) [15], and transformers [16]. Ghaderizadeh et al. [17] proposed a network for hyperspectral image classification that integrates 3D and 2D convolution operations. Three-dimensional (3D) convolution is employed to extract spectral–spatial information, while two-dimensional (2D) convolution further enhances spatial information, avoiding the loss of spectral information. Feng et al. [18] introduced multi-complementary GANs with contrastive learning (CMC-GAN) based on contrastive learning, utilising a network structure incorporating coarse-grained and fine-grained GAN modules to achieve complementary multiscale spatial–spectral features. Zhong et al. [19] presented a hyperspectral image classification algorithm based on a transformer structure consisting of spectral feature extraction and spatial attention modules. Combining these two modules maximises the utilisation of spectral–spatial information in hyperspectral images and enhances the interaction ability among image pixels. He et al. [20] proposed a generative adversarial network with a transformer structure. This model fully attends to global contextual information among pixels through a self-attention mechanism, thereby avoiding the loss of low-level texture information. Yang et al. [21] introduced a ViT [22] structure with convolutional operations into hyperspectral image classification. They introduced the SACP module to capture spectral–spatial fusion information, leading to more accurate classification results. However, deep-learning-based hyperspectral image classification algorithms rely on data-driven methods that require many samples. Zhong et al. [23] introduced an end-to-end spectral–spatial residual ResNet (SSRN) that uses a series of 3D convolutions in the respective residual blocks to extract discriminative joint representation. Roy [24] presents an attention-based adaptive spectral–spatial kernel improved residual network (A2S2K-ResNet) with spectral attention, which learns selective 3D convolutional kernels to jointly extract spectral–spatial features using improved 3D ResBlocks and adopts an efficient feature recalibration (EFR) mechanism to boost the classification performance. In contrast, the need for more hyperspectral image data often leads to the poor performance of trained models in different scene classification tasks. Furthermore, deep-learning methods lack interpretability, which limits their application in specific scenarios.

There are many traditional methods and deep learning methods in hyperspectral image classification. Still, few algorithms combine the two, and poor data compatibility

makes improving the indicators after incorporating them challenging. To address this challenge, Chen et al. [25] introduced SVMFLE, a novel feature line embedding algorithm based on SVM that enables data dimension reduction and enhances the performance of generative adversarial networks in hyperspectral image classification. Kalaiarasi et al. [26] combined the proximal support vector machine (PSVM) algorithm with deep learning methods to develop a new SVM model, ensuring accurate classification outcomes for hyperspectral images. Even with the integration of traditional and deep learning methods in hybrid networks, further improvements are required in the deep learning component or traditional approaches to exploit the spatial–spectral information specific to hyperspectral images fully.

Therefore, this paper takes advantage of the respective advantages of the two, and we only use the powerful feature extraction ability of the deep network while leaving the classification task to the support vector machine module, effectively avoiding confusion during data fusion. Of course, in future research, we will explore the parameter space of the two methods to achieve further data fusion, which has profound research value for hyperspectral image classification. To merge the interpretability of traditional methods with the robust feature extraction capability of deep-learning techniques, this study introduced the orthogonal self-attention module (OSM) and the channel attention module (CAM). These components enable more accurate feature extraction from hyperspectral images using deep residual networks and subsequent classification using the two-step support vector machine (TSSVM) classifier. The critical contributions of the proposed hyperspectral image classification algorithm, which combines the OSM with a two-stage SVM, are as follows:

(1) This paper presents an OSM that performs global self-attention computation along two orthogonal dimensions: image length and width. This module not only enhances the interaction of spatial information within the image but also prevents feature confusion during the training process, thereby improving the generalisation performance of the network. The proposed approach enhances feature representation and improves network robustness by incorporating OSM.

(2) This study introduces a lightweight CAM integrated into a basic residual block. This CAM module dynamically adjusts the channel weights based on the significance of different spectral channels, enabling the network to focus on essential spectral information while reducing interference from redundant information and noise. Incorporating the CAM enhances the quality and robustness of feature representation in the proposed approach, contributing to improved classification performance in hyperspectral image analysis.

(3) This study proposes a TSSVM classifier to integrate and optimise the model by leveraging the discriminant function of the subclassifier. TSSVM is well suited for handling complex datasets and cases with high noise interference and avoids information loss that may arise from traditional SVM methods. Incorporating TSSVM, the proposed approach achieved superior classification performance in hyperspectral image analysis, thereby validating the effectiveness of the method in dealing with challenging scenarios.

## 2. Materials and Methods

### 2.1. Overall Framework

This paper proposes a novel model for hyperspectral image classification using an orthogonal self-attention ResNet and two-step support vector machine (OSANet-TSSVM), a hyperspectral image classification model that combines a residual network with an OSM and an SVM classifier. The overall framework of the proposed model is illustrated in Figure 1. Initially, the image preprocessing stage involved slicing the spatial dimensions of the original hyperspectral image into $P \times P$ sizes, resulting in $P \times P \times C$ image blocks. Subsequently, a $3 \times 3$ convolution operation was applied to perform preliminary feature extraction on each image block while normalising the data for suitable deep network processing. Subsequently, three cascading layer structures were utilised, with multiple

underlying residual blocks within each structure. Each layer structure is composed of three residual blocks, but the internal structure is slightly different. Considering that downsampling the feature map too early may cause detail loss, Layer 1 does not perform downsampling operation. The first convolution stride in L2 and L3 was set to 2, which downsamples the feature map twice and increases the receptive field of the convolution operation, promoting better feature fusion in the network. In addition, the OSM is incorporated after each layer structure, enabling the network to capture spatial global context information three times during forward propagation, thereby highlighting the features of critical pixels in space. Next, the feature maps are transformed into one-dimensional data using fully connected layers to match the data format required by the SVM classifier. Finally, the classifier processed the data to obtain the final output.

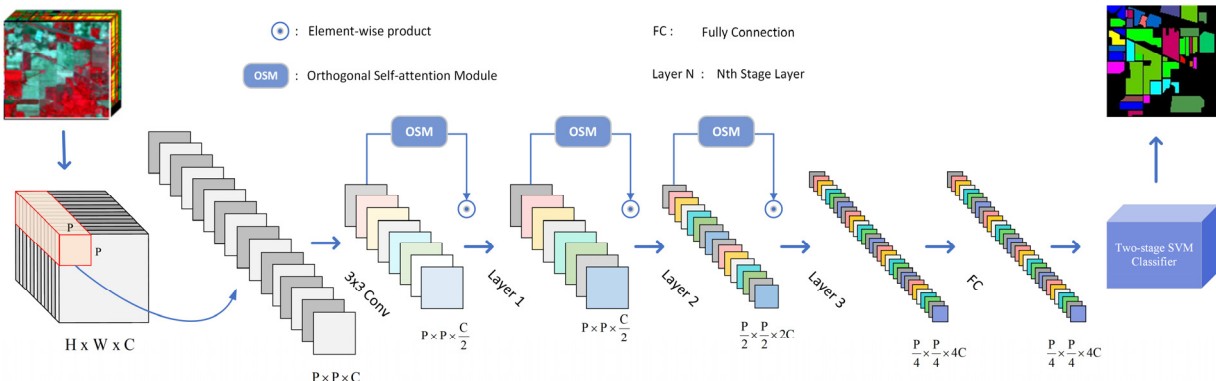

**Figure 1.** The overall framework of OSANet-TSSVM.

## 2.2. Basic Residual Block

The basic residual block serves as the fundamental building block of the residual network, and its detailed implementation is shown in Figure 2. The two convolutional layers in the basic residual block follow a bottleneck structure that initially compresses the feature dimension and then recovers it. This technique effectively reduces the computational cost of convolution while maintaining high performance. Furthermore, the batch normalisation layer standardised the results of each convolution operation to prevent data overflow. The activation layer adds a non-linear expression capability, boosting the network performance. To augment the spatial dimensions of the feature map, the CAM was introduced, positioned before the output of the basic residual block. CAM acts as a channel attention mechanism, dynamically adjusting the channel weights based on the significance of different spectral channels, resulting in improved feature representation and robust classification performance.

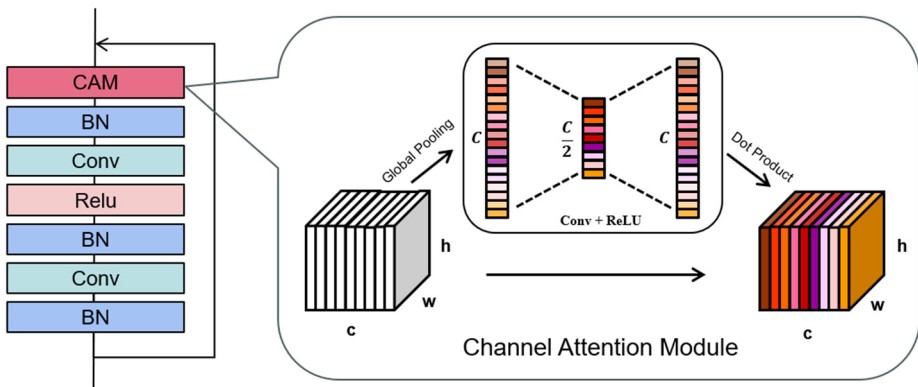

**Figure 2.** Details of basic residual block and channel attention module.

Given that hyperspectral images often consist of numerous spectral bands, it is impractical to input all the bands into the network for feature extraction. Doing so requires substantial computational resources because these bands often exhibit redundant relationships. To address this issue, attention mechanisms are commonly employed to adjust the weight of each band. The spectral attention mechanism, also known as the channel attention mechanism, focuses on extracting features from the spectral dimensions of an image. In this study, a lightweight CAM is integrated into a basic residual block. This integration effectively reduces the computational cost while enabling more detailed weight learning for spectral information. The implementation details are as follows. First, the input undergoes global average pooling to obtain the initial channel-wise weights. Unlike the original channel attention algorithm, global maximum pooling is not employed. This decision is motivated by the fact that such an operation only considers the maximum value of each pixel in the spatial dimension, thereby disregarding the neighbourhood relationship between pixels. Consequently, it is necessary to enhance the weight learning capability and add unnecessary computations. Subsequently, two cascaded convolution layers and normalisation operations perform secondary learning on the attention scores. This process facilitated the acquisition of more accurate spectral weight information.

### 2.3. Orthogonal Self-Attention Module

Deep convolutional neural networks utilise convolutional kernels of varying sizes to extract local features from images. Although widening the kernel size or deepening the network can increase the receptive field, doing so introduces many parameters and increases the image redundancy. Self-attention mechanisms are highly effective for sequence information interaction and parallel computation. Initially applied in natural language processing, these mechanisms have gained widespread use in computer vision in recent years. Visual self-attention mechanisms can search for correlations in an image to obtain global information. However, as the image size increases, the computational complexity of spatial pixel self-attention increases exponentially, severely limiting the application of this mechanism in computer vision. To mitigate the issue of high computational complexity, the OSM is proposed, which utilises depth-wise separable convolution operations to obtain weights and performs self-attention calculations solely within the orthogonal dimensions of image length and width. This approach significantly reduces the number of parameters while effectively capturing the long-range dependencies between pixels in the input image, thereby enabling the extraction of global spatial features.

The implementation details of the OSM are shown in Figure 3. Initially, the input underwent depth-wise separable convolution operations to derive the initial weights Q, K, and V. Subsequently, a reshaping operation was applied to Q and K, followed by element-wise multiplication to generate an attention map with the exact dimensions of the original image width. This attention map is then multiplied by the Reshaped V, yielding the result of the self-attention operation in the width direction. This process is aligned with the original self-attention calculation method. Next, self-attention calculations are conducted in the length direction of the obtained output. By performing these calculations on the orthogonal dimensions of image length and width, the interaction of spatial global information within the image is effectively facilitated.

The computational process of the OSM can be represented as follows:

$$Q, K, V = DWConv(X) \tag{1}$$

$$\text{row-Att}(Q, K, V) = softmax\left(\frac{QK^T}{\alpha}\right)V \tag{2}$$

$$\text{col-Att}(Q, K, V) = softmax\left(\frac{K^T Q}{\alpha}\right)V \tag{3}$$

$$OSA = \text{row-Att}(LN(Col\text{-}Att(X))) + X \tag{4}$$

where *X* denotes the input, DWConv denotes the depth-wise separable convolution operation, *α* represents the scaling factor, *row-Att*(·) and *col-Att*(·) are the components of the OSM in the length and width dimensions, respectively. *LN*(·) indicates the layer normalisation operation and OSA represents the expression for orthogonal self-attention.

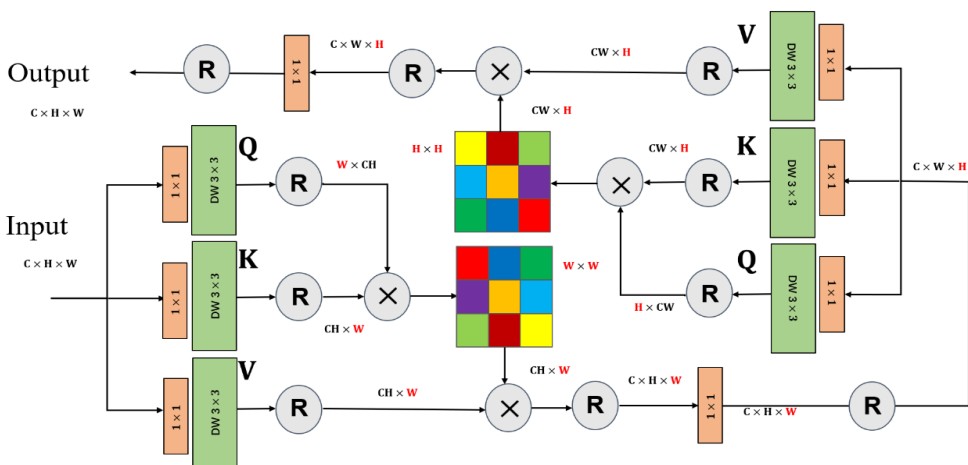

**Figure 3.** Details of orthogonal self-attention module.

## 2.4. Two-Step Support Vector Machine

The working mechanism of SVM can be summarised as follows. It aims to find a classification hyperplane that effectively separates the two classes of sample points in the training set while maximising the margin to this plane. In cases in which classes are linearly inseparable, a kernel function is used to map the data from a low-dimensional input space to a higher-dimensional space. This transformation converts the original linearly inseparable problem in a low-dimensional space into a linearly separable problem in a higher-dimensional space. SVM have a solid theoretical foundation and strong generalisation ability, providing unique advantages in addressing small-sample, non-linear, and high-dimensional pattern recognition problems. The combination of hyperspectral technology and SVM for qualitative substance classification has been successfully demonstrated and applied to various research fields.

A traditional SVM constructs a classifier for binary classification problems to obtain discriminant results. However, multi-class classification problems are typically transformed into multiple binary classification problems. A series of binary classifiers are constructed to obtain discriminant results, commonly employing One-vs-One or One-vs-All classifiers. The final classification is determined by voting or the maximum probability. Although this discriminant approach has been used for decades, outliers and subclassifiers can easily influence the results. In recent years, researchers have focused on constructing kernel functions and optimising various algorithms to enhance the performance of SVM, leading to notable advancements in theory and applications. However, challenges and limitations persist. Because the SVM involves solving a quadratic programming problem, the computation time and memory requirements significantly increase with larger datasets, thereby posing challenges in handling large-scale datasets. Therefore, effectively addressing large-scale data sets remains an active area of research. In addition, noise and outliers may cause model overfitting during the training process, thereby affecting the accuracy of the classification results. Consequently, improving the robustness of SVMs to enhance the noise and outlier tolerance is a significant problem that requires attention.

To overcome these challenges, this paper proposes a novel decision method for multi-class SVM classification. In the first phase of the SVM, multiple subclassifiers are trained, and a second phase of training is introduced to address the limitations of the individual classifiers. The discriminant results obtained by the one-step subclassifiers were retained and remapped as new features. A second SVM training was performed on the newly

established mappings, resulting in the TSSVM. By integrating the results of multiple classifiers, this approach compensates for the shortcomings of individual classifiers and allows complex classification problems to be solved effectively. Figure 4 illustrates the research roadmap adopted in this study.

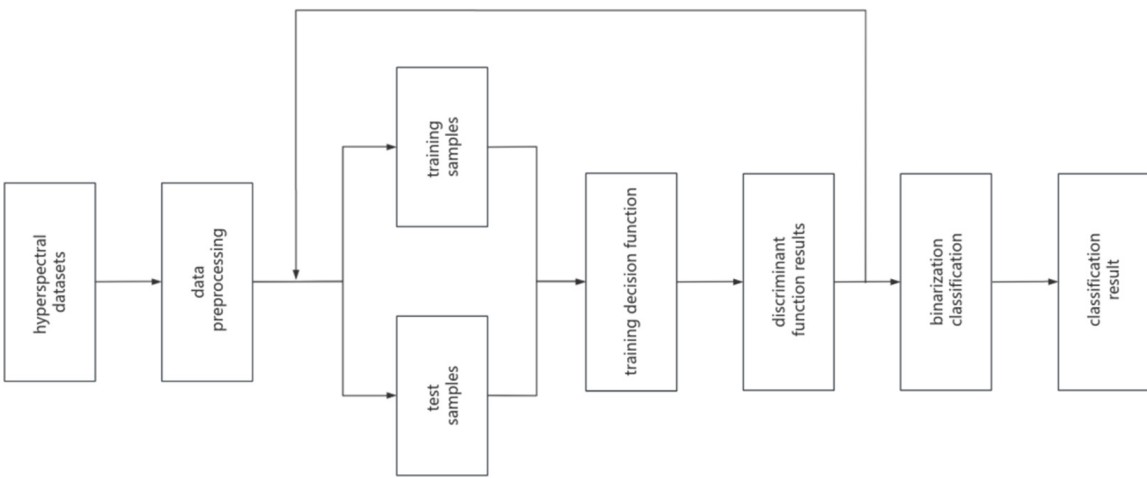

**Figure 4.** The classification process of TSSVM.

Given the original training samples $x_i$, $i = 1, 2, \ldots, k$ with N classes and M sub-discriminators, M discriminant functions $f_1(\cdot)$, $f_2(\cdot), \ldots, f_M(\cdot)$ without binarisation are obtained through the one-step SVM training using the training samples and their class labels. For instance, in One-vs-All classifiers, a 0–1 value is obtained by determining the class attribution of unknown samples, representting the possibility of the test sample being in that class. The traditional approach involves inputting an unknown sample into each of the M subclassifiers to obtain M discriminant results, taking the maximum value as the highest possibility in that subclassifier, as listed in Table 1.

**Table 1.** One-vs-All one-step SVM classification discrimination.

| Hyperspectral Feature Vector | Sub-Classifier $f_1$ | Sub-Classifier $f_2$ | Sub-Classifier $f_3$ | ... | Sub-Classifier $f_M$ | Classification |
|---|---|---|---|---|---|---|
| $x_1$ | $f_1(x_1)$ | $f_2(x_1)$ | $f_3(x_1)$ | ... | $f_M(x_1)$ | Max |
| $x_2$ | $f_1(x_2)$ | $f_2(x_2)$ | $f_3(x_2)$ | ... | $f_M(x_2)$ | Max |
| ... | ... | ... | ... | ... | ... | ... |
| $x_k$ | $f_1(x_k)$ | $f_2(x_k)$ | $f_3(x_k)$ | ... | $f_M(x_k)$ | Max |

Although the traditional SVM method is simple and computationally efficient, its scalability and classification performance become more robust, especially in cases of overlapping categories or when non-linear separability is present. To address these limitations, the TSSVM model concatenates multiple SVMs and fully utilises the advantages of various subclassifiers to solve complex classification problems. Specifically, the M discriminant functions obtained from the one-step SVM training were retained as new feature vectors $\widetilde{x}_i = (f_1(x_i), f_2(x_i), \ldots, f_m(x_i))$, $i = 1, 2, \ldots, k$. Using the remapped feature vectors and original label data as training data, M two-step sub-classifiers are trained and obtained, denoted as $df_1(\cdot)$, $df_2(\cdot)$, $df_3(\cdot)$, $df_4(\cdot)$. The two-step subclassifiers determine the result vector of the one-step subclassifiers and consider the maximum value as the result of the two-step subclassifier, as listed in Table 2. Each sub-classifier obtained from the one-step training process had different features or data subsets. Simultaneously, TSSVM integrates and optimises the discriminant results of the subclassifiers to obtain more accurate classification results.

**Table 2.** One-vs-All two-step SVM classification discrimination.

| Hyperspectral Feature Vector | Sub-Classifier $df_1$ | Sub-Classifier $df_2$ | Sub-Classifier $df_3$ | $\cdots$ | Sub-Classifier $df_4$ |
|---|---|---|---|---|---|
| $\widetilde{x}_1$ | $df_1(\widetilde{x}_1)$ | $df_2(\widetilde{x}_1)$ | $df_3(\widetilde{x}_1)$ | $\cdots$ | $df_4(\widetilde{x}_1)$ |
| $\widetilde{x}_2$ | $df_1(\widetilde{x}_2)$ | $df_2(\widetilde{x}_2)$ | $df_3(\widetilde{x}_2)$ | $\cdots$ | $df_4(\widetilde{x}_2)$ |
| $\cdots$ | $\cdots$ | $\cdots$ | $\cdots$ | $\cdots$ | $\cdots$ |
| $\widetilde{x}_k$ | $df_1(\widetilde{x}_k)$ | $df_2(\widetilde{x}_k)$ | $df_3(\widetilde{x}_k)$ | $\cdots$ | $df_4(\widetilde{x}_k)$ |

The TSSVM model proposed in this study optimised its classification performance through secondary training of the discriminant functions obtained from the one-step SVM training. It then makes decisions by voting or probability weighting. By integrating the results of multiple classifiers, the effects of noise and outliers on the final decision can be reduced, thereby improving the accuracy and robustness of classification.

## 3. Experimental Analysis

The experiments were conducted on a computer with an Intel i7 processor and 32 GB of memory. The algorithm model was implemented using Python 3.7 and developed in the PyCharm integrated development environment. A pyramid residual network model was employed to construct a deep backbone network, which was implemented using the PyTorch framework. The training was performed for 150 epochs on an NVIDIA GeForce RTX 4090 GPU with a batch size of 200 epochs. The SGD optimiser with an initial learning rate of 0.1 was utilised. The manufacturer of the GPU was NVIDIA and the server motherboard generator was ASUS, both made in China. The SVM classification algorithm was trained using a nonlinear Gaussian kernel function with 10-fold cross-validation. A non-linear Gaussian kernel function was employed for the SVM classification algorithm, and training was conducted using a 10-fold cross-validation.

This section presents the experimental results obtained from the Indian Pine, Salinas, Pavia University, and Kennedy Space Center datasets. The classification results of seven models, namely SVM, AENSVM, TSSVM, ResNet, SSRN [23], A2S2K-ResNet [24], and OSANet-TSSVM, were compared. The experimental results demonstrate that the proposed OSANet-TSSVM model performs better in terms of subjective and objective evaluation metrics.

### 3.1. Evaluating Indicator

A confusion matrix is an essential tool for evaluating the performance of the classification models. This provides detailed information on the classification results of the model for each class. Utilising the confusion matrix, the performance of the classification model for different classes can be intuitively observed. Its form is:

$$M = \begin{bmatrix} P_{11} & P_{12} & & P_{1n} \\ & & \cdots & \\ P_{21} & P_{22} & & P_{2n} \\ & \vdots & \ddots & \vdots \\ P_{n1} & P_{n2} & \cdots & P_{nn} \end{bmatrix} \tag{5}$$

The overall accuracy (OA), average accuracy (AA), and Kappa coefficient (k) were calculated using a confusion matrix to evaluate the overall performance of the model comprehensively. The OA represents the proportion of correctly classified samples to the total number of samples in the entire dataset. AA represents the average accuracy rate for each class, serving as a more comprehensive metric by accounting for variations among different classes and evaluating the classification ability of the model across each class. The Kappa coefficient is a statistical measure of the agreement between the classification model and random chance. It considers the differences between the classification accuracy of the

model and the accuracy of the random selection. These three evaluation metrics are defined in Formulas (6)–(8), where $n$ represents the number of classes and the number of samples of class $i$ is predicted as class $j$.

$$OA = \frac{\sum_{i=1}^{n} P_{ii}}{\sum_{i=1}^{n} \sum_{j=1}^{n} P_{ij}} \tag{6}$$

$$AA = \frac{\sum_{i=1}^{n} \frac{P_{ii}}{\sum_{j=1}^{n} P_{ij}}}{n} \tag{7}$$

$$Kappa = \frac{\sum_{i=1}^{n} P_{ii} - E}{\sum_{i=1}^{n} \sum_{j=1}^{n} P_{ij} - E}$$
$$E = \frac{\left(\sum_{i=1}^{n} \sum_{j=1}^{n} P_{ij}\right) \cdot \left(\sum_{i=1}^{n} \sum_{j=1}^{n} P_{ji}\right)}{\left(\sum_{i=1}^{n} \sum_{j=1}^{n} P_{ij}\right)^2} \tag{8}$$

### 3.2. Indian Pines Dataset

The Indian Pines dataset was collected using the AVIRIS sensor over the state of Indiana in the central northern region of the United States. The image size was $144 \times 144$ pixels with 224 spectral bands, a spatial resolution of 20 m, and a wavelength range of 400–500 nm. After removing the 4 zero bands and 20 water absorption bands, the remaining 200 bands were used for the experiments. The Indian Pines dataset consists of 10,249 ground objects labelled into 16 classes. Due to significant variations in sample sizes among different classes in this dataset, 10% of the samples from each class were randomly selected as training data, while the remaining 90% constituted the test set. Figure 5 shows the corresponding high-resolution pseudocolour image and ground truth map. Table 3 lists the number of samples in the training and test sets used in the experiments.

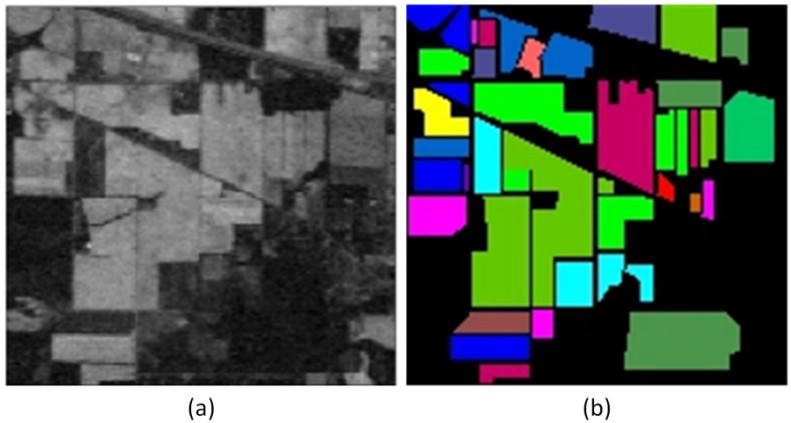

(a)  (b)

**Figure 5.** Indian Pines dataset. (**a**) Pseudocolour image for HSI. (**b**) Ground truth map.

**Table 3.** Number of samples, test set, and training set for each class in the Indian Pines dataset.

| No. | Class | Sample | Training | Testing |
|---|---|---|---|---|
| 1 | Alfalfa | 54 | 5 | 49 |
| 2 | Corn-notill | 1434 | 143 | 1291 |
| 3 | Corn-mintill | 834 | 83 | 751 |
| 4 | Corn | 234 | 24 | 210 |
| 5 | Grass-pasture | 497 | 48 | 449 |
| 6 | Grass-trees | 747 | 73 | 674 |
| 7 | Grass-pasture-mowed | 26 | 3 | 23 |
| 8 | Hay-windrowed | 489 | 48 | 441 |
| 9 | Oats | 20 | 2 | 18 |

**Table 3.** *Cont.*

| No. | Class | Sample | Training | Testing |
|:---:|:---:|:---:|:---:|:---:|
| 10 | Soybean-notill | 968 | 97 | 871 |
| 11 | Soybean-mintill | 2468 | 245 | 2223 |
| 12 | Soybean-clean | 614 | 59 | 555 |
| 13 | Wheat | 212 | 20 | 192 |
| 14 | Woods | 1294 | 126 | 1168 |
| 15 | Buildings-Grass-Trees-Drives | 300 | 39 | 261 |
| 16 | Stone-Steel-Towers | 95 | 9 | 86 |
| | Total | 10,286 | 1024 | 9262 |

A well-divided training set was used to train the classification models: SVM, AENSVM, TSSVM, ResNet, SSRN, A2S2K-ResNet, and OSANet-TSSVM. Table 4 presents the overall accuracy (OA), average accuracy (AA), and Kappa coefficient for each class across different models. The highest accuracy values are highlighted in bold. Figure 6 presents a visual comparison of different classification models.

**Table 4.** Classification results of different algorithms on Indian Pines dataset.

| Class No. | SVM | AENSVM | TSSVM | ResNet | SSRN | A2S2K-ResNet | OSANet-SVM |
|:---:|:---:|:---:|:---:|:---:|:---:|:---:|:---:|
| 1 | 77.78 | 80.95 | 74.19 | 80.49 | 77.35 | 95.12 | **97.56** |
| 2 | 76.55 | 75.35 | 80.73 | 88.02 | 93.33 | 97.35 | **97.90** |
| 3 | 76.65 | 80.17 | 77.68 | 83.53 | 95.66 | 97.52 | **98.26** |
| 4 | 64.62 | 68.90 | 66.49 | 60.09 | 92.56 | 88.73 | **97.65** |
| 5 | 93.58 | 95.99 | 94.68 | 88.28 | 92.27 | 99.21 | **99.31** |
| 6 | 95.37 | 91.06 | 94.33 | 92.09 | 94.60 | **99.85** | 99.54 |
| 7 | 60.00 | 86.67 | 77.78 | 36.00 | 55.56 | 80.00 | **92.00** |
| 8 | 94.16 | 97.73 | 93.03 | 98.84 | 99.77 | 99.62 | **99.81** |
| 9 | 52.63 | 47.62 | 71.43 | 5.56 | 80.00 | 88.89 | **100.00** |
| 10 | 71.64 | 74.33 | 77.97 | 78.29 | 95.21 | 95.77 | **97.49** |
| 11 | 79.01 | 86.05 | 82.67 | 88.33 | 96.40 | 99.00 | **99.28** |
| 12 | 81.97 | 84.84 | 86.18 | 66.29 | 92.57 | 92.13 | **94.76** |
| 13 | 95.26 | 93.97 | 95.88 | 92.43 | 92.06 | **100.00** | **100.00** |
| 14 | 90.95 | 93.70 | 93.99 | 92.36 | 99.82 | 98.07 | **98.42** |
| 15 | 79.30 | 69.47 | 78.67 | 74.64 | 89.82 | **97.12** | 97.05 |
| 16 | 93.33 | 98.70 | 95.71 | 70.24 | 92.42 | **97.62** | 96.43 |
| OA (%) | 82.05 | 84.33 | 84.57 | 85.33 | 96.24 | 97.53 | **98.19** |
| AA (%) | 82.44 | 82.84 | 83.29 | 80.21 | 86.94 | 97.11 | **97.73** |
| K × 100 | 79.46 | 82.14 | 82.94 | 83.26 | 94.11 | 97.24 | **98.13** |

The results show that the classification effect of the improved AENSVM and TSSVM is improved compared with the traditional SVM. Still, these conventional methods only perform simple spectral feature processing, which can easily lead to the misclassification phenomenon. The proposed method achieved the best classification accuracy with an OA of 98.19%. For the three comprehensive evaluation indicators, AA, OA, and Kappa coefficient of the Indian Pines dataset, the proposed method outperformed the second-best A2S2K-ResNet method by 0.65%, 0.62%, and 0.89%, respectively. In classes with fewer training samples, such as alfalfa, corn, grass-pasture-mowed, and oats, the classification accuracy significantly improved to 97.56%, 97.65%, 92.00%, and 99.81%, respectively. Figure 6 shows that, compared with other models, the OSANet-TSSVM and A2S2K-ResNet models appear to have better classification performance, overcoming the problem of smooth margin transition.

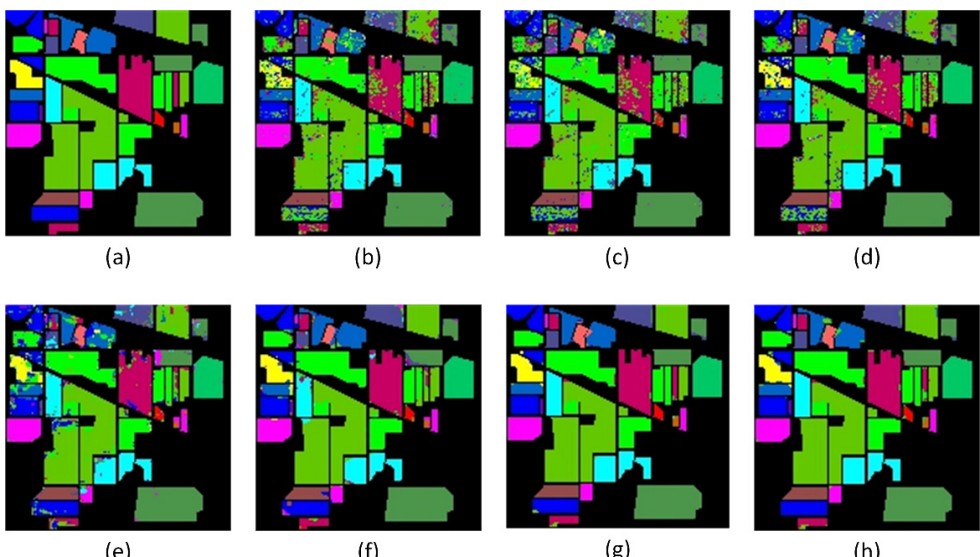

**Figure 6.** Indian Pines dataset. (**a**) Ground truth; (**b**) SVM; (**c**) AENSVM; (**d**) TSSVM; (**e**) ResNet; (**f**) SSRN; (**g**) A2S2K-ResNet; and (**h**) OSANet-TSSVM.

Moreover, the OSANet-TSSVM model exhibited significantly better classification performance than the other models in the alfalfa and soybean-no-till classes, with more apparent visual boundary contours closer to the ground truth. This indicates that the proposed OSANet-TSSVM can help the model better capture the relationships between different bands and positions, leading to more accurate image classification. Not only did this improve the overall accuracy, but the generator also generated diverse samples in classes with few training samples, learning the essential features of these samples.

### 3.3. Pavia University Dataset

The Pavia University dataset is a hyperspectral remote sensing image dataset obtained from Pavia, Italy, using a ROSIS sensor from Germany. It comprises 115 bands covering wavelengths from 0.43 to 0.86 μm. After filtering out the noisy bands, the dataset comprised 102 bands with a spatial resolution of 1.3 m. The image dimensions were 610 × 340 pixels, totalling 42,776 pixels. In addition to the background, the dataset contained nine land cover classes, including residential areas, grasslands, and trees. Because of its high spatial resolution and diverse urban landscapes, it has gained popularity in hyperspectral remote sensing image classification research.

In this experiment, a subset of 100 samples per class was randomly selected for training, and the remaining samples were used for testing. The detailed quantities of the samples for each class, training, and test sample are presented in Table 5. Figure 7 shows the pseudocolour images and the overall sample label classification map.

**Table 5.** Number of samples, test set, and training set for each class in the Pavia University dataset.

| No. | | Class | Sample | Training | Testing |
|---|---|---|---|---|---|
| 1 | | Asphalt | 6631 | 100 | 6531 |
| 2 | | Meadows | 18,649 | 100 | 18,549 |
| 3 | | Gravel | 2099 | 100 | 1999 |
| 4 | | Trees | 3064 | 100 | 2964 |
| 5 | | Painted metal sheets | 1345 | 100 | 1245 |
| 6 | | Bare Soil | 5029 | 100 | 4929 |
| 7 | | Bitumen | 1330 | 100 | 1230 |
| 8 | | Self-Blocking Bricks | 3682 | 100 | 3582 |
| 9 | | Shadows | 947 | 100 | 847 |
| | | Total | 42,776 | 900 | 41,876 |

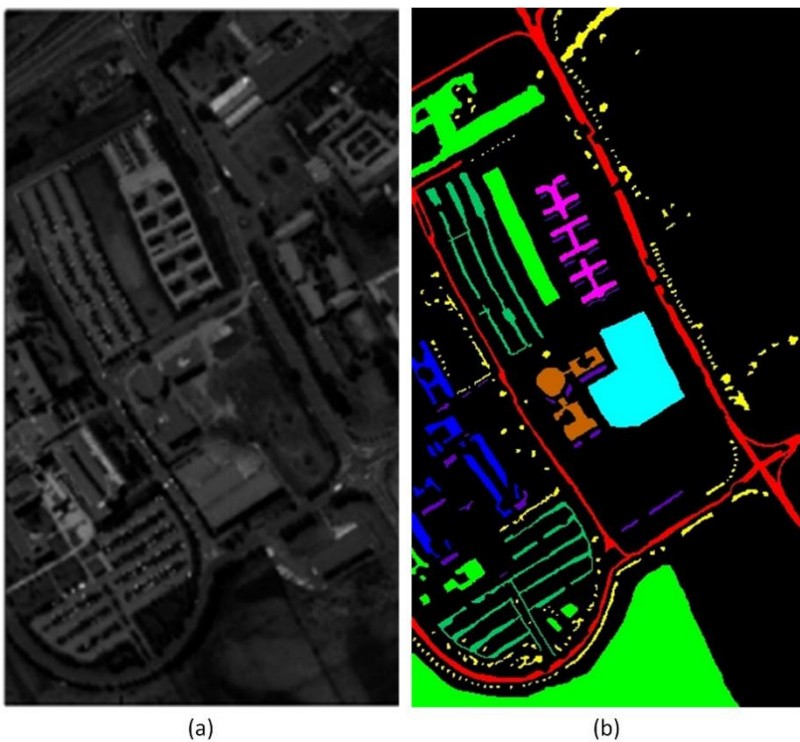

**Figure 7.** Pavia University dataset. (**a**) Pseudocolour image for HSI. (**b**) Ground truth map.

Table 6 presents the results of the OSANet-TSSVM and six other comparative algorithms. Along with overall accuracy (OA), average accuracy (AA), and Kappa values, the table displays the accuracy values for each class. The highest precision values are highlighted in bold text in the Experimental Results section. Figure 8 illustrates the classification results of the seven methods on the Pavia University dataset.

**Table 6.** Classification results of different algorithms on Pavia University dataset.

| Class No. | SVM | AENSVM | TSSVM | ResNet | SSRN | A2S2K-ResNet | OSANet-SVM |
|---|---|---|---|---|---|---|---|
| 1 | 96.62 | 94.67 | 96.27 | 91.09 | **99.54** | 98.79 | 99.49 |
| 2 | 96.66 | 96.59 | 97.40 | 92.09 | 99.36 | 98.89 | **99.56** |
| 3 | 65.22 | 73.27 | 67.49 | 59.18 | 83.67 | **98.95** | 95.65 |
| 4 | 77.94 | 90.15 | 84.14 | 95.88 | 99.20 | 97.50 | **99.33** |
| 5 | 96.40 | 99.68 | 99.30 | **100.00** | 100.00 | 100.00 | 100.00 |
| 6 | 68.54 | 78.32 | 75.85 | 73.16 | 97.66 | 98.23 | **98.35** |
| 7 | 59.48 | 70.70 | 60.28 | 63.09 | 93.23 | **98.37** | 95.25 |
| 8 | 76.08 | 87.20 | 78.60 | 89.92 | 96.59 | 95.73 | **95.77** |
| 9 | **100.00** | **100.00** | **100.00** | 97.40 | **100.00** | 99.65 | 99.60 |
| OA (%) | 85.22 | 90.52 | 88.14 | 87.74 | 97.17 | 98.47 | **98.64** |
| AA (%) | 89.34 | 87.84 | 91.04 | 87.88 | 97.67 | 98.01 | **98.34** |
| K × 100 | 80.89 | 87.48 | 84.40 | 83.57 | 97.29 | 97.95 | **98.17** |

The analysis in Table 6 reveals that OSANet-TSSVM achieved the best classification results on the Pavia University dataset, demonstrating significant advantages over traditional algorithms and residual neural networks. It slightly outperformed the second-best A2S2K-ResNet method with OA, AA, and Kappa values of 0.17%, 0.33%, and 0.22%, respectively. Furthermore, the OSANet-TSSVM excels in meadows, trees, painted metal sheets, bare soil, and self-blocking rice classes. The results also show that the improved AEN-SVM and TSSVM methods based on traditional SVM have surpassed ResNet regarding classification performance. Specifically, AEN-SVM improved by 2.78% in the OA metric,

while TSSVM attained an AA metric of 91.04%. This suggests that SVM continues to exhibit great potential for hyperspectral image classification. Figure 8 presents the classification results of the seven methods on the Pavia University dataset, showing OSANet-TSSVM with the fewest misclassified labels aligned closer to the ground truth. This demonstrates that the OSANet-TSSVM prioritises crucial spectral information and enhances spectral feature extraction capabilities.

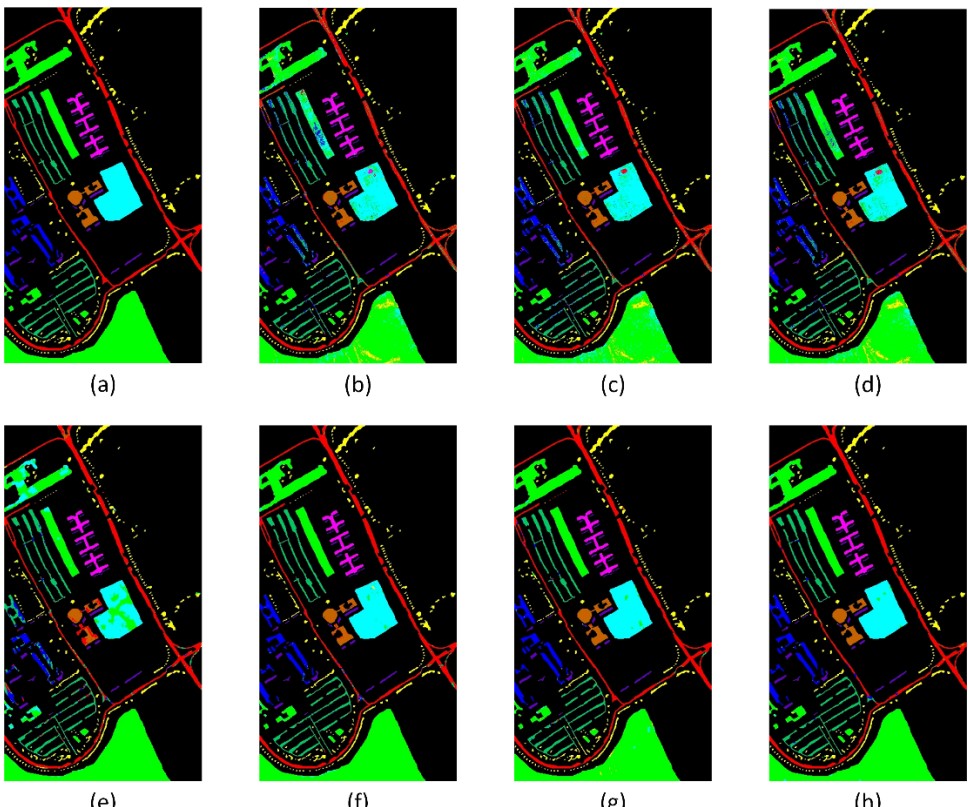

**Figure 8.** Pavia University dataset. (**a**) Ground truth; (**b**) SVM; (**c**) AENSVM; (**d**) TSSVM; (**e**) ResNet; (**f**) SSRN; (**g**) A2S2K-ResNet; and (**h**) OSANet-TSSVM.

### 3.4. Salinas Dataset

The Salinas dataset is a hyperspectral remote sensing image dataset from Salinas Valley, California, USA. Collected using NASA's AVIRIS sensor, it comprises 224 spectral channels with a spatial resolution of 3.7 m and an image size of $512 \times 217$ pixels. The dataset includes 16 land cover classes, including various vegetables, orchards, and vineyards, making it a challenging dataset for hyperspectral image classification and target detection. Owing to its high resolution and complex land cover types, the Salinas dataset has become an essential benchmark in this field. The land cover information is presented in Table 7, and Figure 9 depicts a band image and ground truth map from the Salinas dataset.

**Table 7.** Number of samples, test set, and training set for each class in the Salinas dataset.

| No. | | Class | Sample | Training | Testing |
|---|---|---|---|---|---|
| 1 | | Brocoli_green_weeds_1 | 2009 | 100 | 1909 |
| 2 | | Brocoli_green_weeds_2 | 3726 | 100 | 3626 |
| 3 | | Fallow | 1976 | 100 | 1876 |
| 4 | | Fallow_rough_plow | 1394 | 100 | 1294 |
| 5 | | Fallow_smooth | 2678 | 100 | 2578 |
| 6 | | Stubble | 3959 | 100 | 3859 |
| 7 | | Celery | 3579 | 100 | 3479 |

**Table 7.** *Cont.*

| No. | Class | Sample | Training | Testing |
|---|---|---|---|---|
| 8 | Grapes_untrained | 11,271 | 100 | 11,171 |
| 9 | Soil_vinyard_develop | 6203 | 100 | 6103 |
| 10 | Corn_senesced_green_weeds | 3278 | 100 | 3178 |
| 11 | Lettuce_romaine_4wk | 1068 | 100 | 968 |
| 12 | Lettuce_romaine_5wk | 1927 | 100 | 1827 |
| 13 | Lettuce_romaine_6wk | 916 | 100 | 816 |
| 14 | Lettuce_romaine_7wk | 1070 | 100 | 970 |
| 15 | Vinyard_untrained | 7268 | 100 | 7168 |
| 16 | Vinyard_vertical_trellis | 1807 | 100 | 1707 |
| | Total | 54,129 | 1600 | 52,529 |

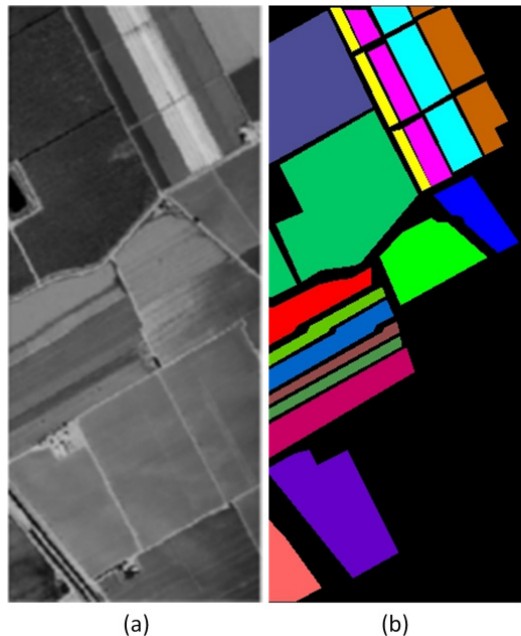

(a)  (b)

**Figure 9.** Salinas dataset. (**a**) Pseudocolour image for HSI. (**b**) Ground truth map.

Table 8 provides a detailed comparison of the experimental accuracy between OSANet-TSSVM and the six other classification models. The highest accuracy values are highlighted in bold. Figure 10 shows the classification results listed in Table 8.

**Table 8.** Classification results of different algorithms on the Salinas dataset.

| Class No. | SVM | AENSVM | TSSVM | ResNet | SSRN | A2S2K-ResNet | OSANet-SVM |
|---|---|---|---|---|---|---|---|
| 1 | 99.95 | **100.00** | **99.84** | 89.79 | **100.00** | **100.00** | 99.54 |
| 2 | 99.42 | 99.94 | 99.37 | 99.45 | 99.12 | **100.00** | **100.00** |
| 3 | 99.21 | 99.05 | 99.47 | 99.95 | 96.12 | 98.42 | **99.81** |
| 4 | 98.31 | 98.32 | 98.54 | 97.91 | 96.41 | 99.01 | 99.82 |
| 5 | 96.33 | 98.73 | 97.11 | 97.40 | 99.80 | 99.78 | **100.00** |
| 6 | 99.74 | **100.00** | **99.95** | 98.55 | **100.00** | **100.00** | **100.00** |
| 7 | 99.91 | 99.68 | 99.88 | 98.79 | 99.43 | 99.56 | **100.00** |
| 8 | 82.05 | 79.45 | 82.46 | 87.14 | 92.28 | 94.18 | **96.98** |
| 9 | 99.36 | 98.99 | 98.92 | 99.59 | 99.00 | **100.00** | **100.00** |
| 10 | 92.71 | 95.19 | 93.59 | 94.65 | 99.47 | 98.35 | **99.97** |
| 11 | 97.96 | 96.98 | 98.07 | 99.25 | 94.42 | 98.67 | **99.87** |
| 12 | 98.38 | 99.24 | 98.97 | 99.51 | 99.84 | 98.32 | **100.00** |
| 13 | 99.75 | 99.26 | 99.75 | **100.00** | 99.15 | **100.00** | **100.00** |

**Table 8.** *Cont.*

| Class No. | SVM | AENSVM | TSSVM | ResNet | SSRN | A2S2K-ResNet | OSANet-SVM |
|---|---|---|---|---|---|---|---|
| 14 | 90.57 | 94.65 | 92.57 | 98.35 | 98.48 | 99.79 | **99.94** |
| 15 | 66.18 | 68.64 | 69.45 | 80.25 | 90.60 | 93.12 | **97.96** |
| 16 | 98.19 | 99.29 | 98.82 | 99.24 | 99.13 | 99.30 | **99.73** |
| OA (%) | 90.21 | 90.63 | 91.03 | 93.40 | 95.49 | 97.73 | **98.32** |
| AA (%) | 95.90 | 95.46 | 96.16 | 95.36 | 97.21 | 97.45 | **99.29** |
| K × 100 | 89.10 | 89.54 | 90.00 | 92.58 | 96.08 | 98.71 | **98.11** |

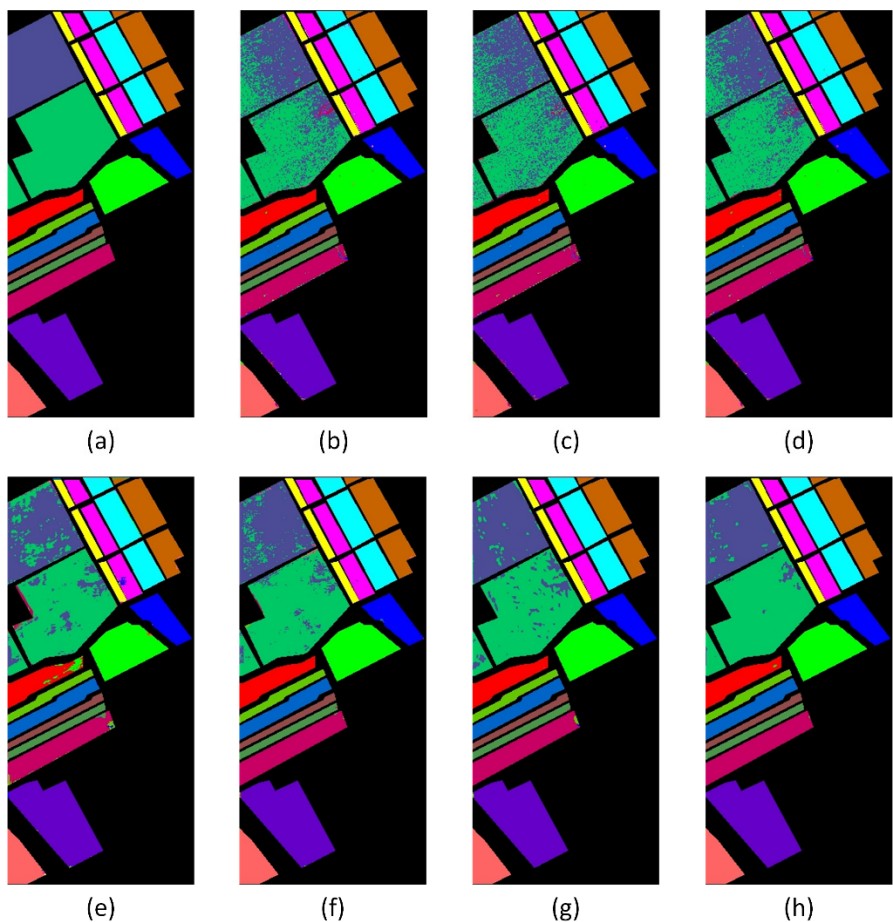

**Figure 10.** Salinas dataset. (**a**) Ground truth; (**b**) SVM; (**c**) AENSVM; (**d**) TSSVM; (**e**) ResNet; (**f**) SSRN; (**g**) A2S2K-ResNet; and (**h**) OSANet-TSSVM.

Table 8 shows that compared to other methods, the proposed OSANet-TSSVM achieves the highest classification performance, with an OA of 98.32%, an AA of 99.29%, and a Kappa coefficient of 98.11%. These results indicate that OSANet-TSSVM achieves the highest accuracy in most categories and reaches 100% classification accuracy in seven categories. In contrast, the remaining algorithms struggle to accurately classify the Grapes_untrained and Vinyard_untrained classes, while the proposed model significantly improves the accuracy, surpassing the second-best A2S2K-ResNet model by 2.80% and 4.84%, respectively. Additionally, comparison images demonstrate that the experimental model exhibits good classification performance, especially in the Grapes_untrained, Vinyard_untrained, and Corn_senesced_green_weeds classes, with fewer misclassifications. These results confirm the effectiveness of the proposed OSANet-TSSVM in extracting spectral feature information and achieving more accurate feature classification.

### 3.5. Kennedy Space Center Dataset

The Kennedy Space Center (KSC) dataset is a hyperspectral remote sensing image dataset collected by NASA's AVIRIS sensor at the Kennedy Space Center in the United States. This dataset contains 224 spectral bands with a spatial resolution of 18 m. After removing the water absorption bands and low signal-to-noise ratio bands, the actual bands used for training are 176. The image size is 512 × 614 pixels, covering 13 land cover categories such as wetlands, woodland, and beaches. The KSC dataset is widely used in hyperspectral remote sensing image classification and ecological research due to its rich wetland ecosystem and complex surface characteristics. Table 9 shows the ground object category information, and Figure 11 shows a certain band image and ground truth map from the KSC dataset.

**Table 9.** Number of samples, test set, and training set for each class in the KSC dataset.

| No. | Class | Sample | Training | Testing |
|---|---|---|---|---|
| 1 | Scrub | 761 | 76 | 685 |
| 2 | Willow swamp | 243 | 24 | 219 |
| 3 | CP hammock | 256 | 26 | 230 |
| 4 | Slash pine | 252 | 25 | 227 |
| 5 | Oak/Broadleaf | 161 | 16 | 145 |
| 6 | Hardwood | 229 | 23 | 206 |
| 7 | Swamp | 105 | 11 | 94 |
| 8 | Graminoid marsh | 431 | 43 | 388 |
| 9 | Spartina marsh | 520 | 52 | 468 |
| 10 | Cattail marsh | 404 | 40 | 364 |
| 11 | Salt marsh | 419 | 42 | 377 |
| 12 | Mud flats | 503 | 50 | 453 |
| 13 | Water | 927 | 93 | 834 |
| | Total | 5211 | 521 | 4690 |

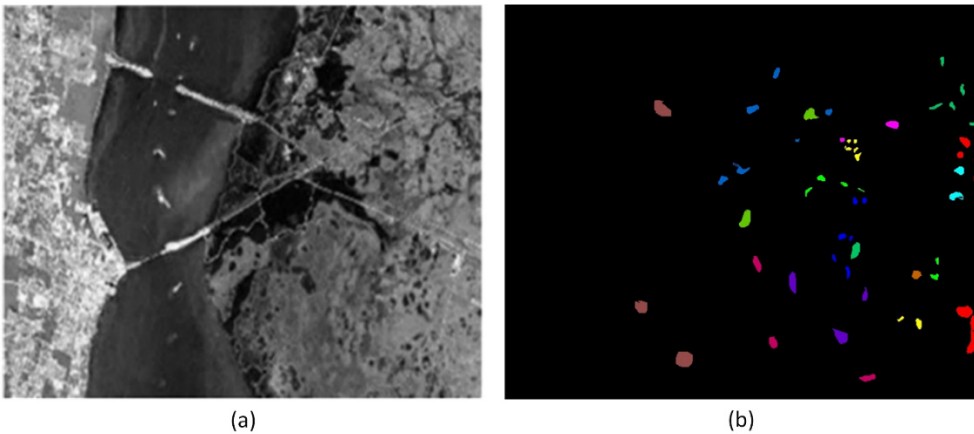

(a)                                   (b)

**Figure 11.** KSC dataset. (**a**) Pseudocolour image for HSI. (**b**) Ground truth map.

Table 10 presents a detailed comparison of the experimental accuracy of the OSANet-SVM model against other classification models, including random forest (RF), support vector machine (SVM), AEN-SVM, ResNet, SSRN, and A2S2K-ResNet. The highest accuracy values among the compared models are highlighted in bold text. Figure 12 sequentially displays the recovery maps obtained from the classification of the RF, SVM, AEN-SVM, ResNet, SSRN, and A2S2K-ResNet models, corresponding to the detailed classification results presented in Table 10.

**Table 10.** Classification results of different algorithms on the KSC dataset.

| Class No. | SVM | AENSVM | TSSVM | ResNet | SSRN | A2S2K-ResNet | OSANet-SVM |
|---|---|---|---|---|---|---|---|
| 1 | 95.47 | 93.66 | 95.82 | 99.85 | 96.24 | **100.00** | 99.62 |
| 2 | 93.02 | 91.36 | 95.19 | 68.04 | 95.81 | 97.72 | **100.00** |
| 3 | 90.34 | 82.26 | 87.67 | 53.04 | 88.89 | 90.52 | **98.32** |
| 4 | 71.23 | 70.71 | 76.89 | 42.73 | **86.09** | 72.69 | 84.66 |
| 5 | 76.03 | 81.13 | 75.94 | 20.69 | 80.80 | 93.81 | **96.55** |
| 6 | 72.12 | 86.98 | 82.08 | 97.57 | 89.33 | **100.00** | 98.12 |
| 7 | 85.26 | 86.52 | 84.62 | 96.81 | 86.67 | 89.36 | **100.00** |
| 8 | 94.43 | 92.13 | 94.52 | 79.12 | 94.12 | 98.20 | **99.67** |
| 9 | 94.07 | 92.61 | 96.24 | 99.36 | 96.24 | 99.23 | **99.73** |
| 10 | 96.69 | 96.03 | 97.81 | 92.03 | 94.92 | **99.24** | 100.00 |
| 11 | 95.32 | 99.73 | 96.44 | 100.00 | 99.65 | 100.00 | 100.00 |
| 12 | 97.44 | 94.04 | 97.47 | 89.18 | 98.53 | 100.00 | 100.00 |
| 13 | 99.05 | **100.00** | 99.46 | **100.00** | 99.54 | 100.00 | 100.00 |
| OA (%) | 92.80 | 92.65 | 93.09 | 87.22 | 95.19 | 98.45 | **98.75** |
| AA (%) | 88.91 | 89.78 | 90.23 | 80.21 | 92.91 | 97.43 | **97.94** |
| K × 100 | 91.99 | 91.81 | 92.42 | 85.89 | 94.65 | 97.15 | **98.63** |

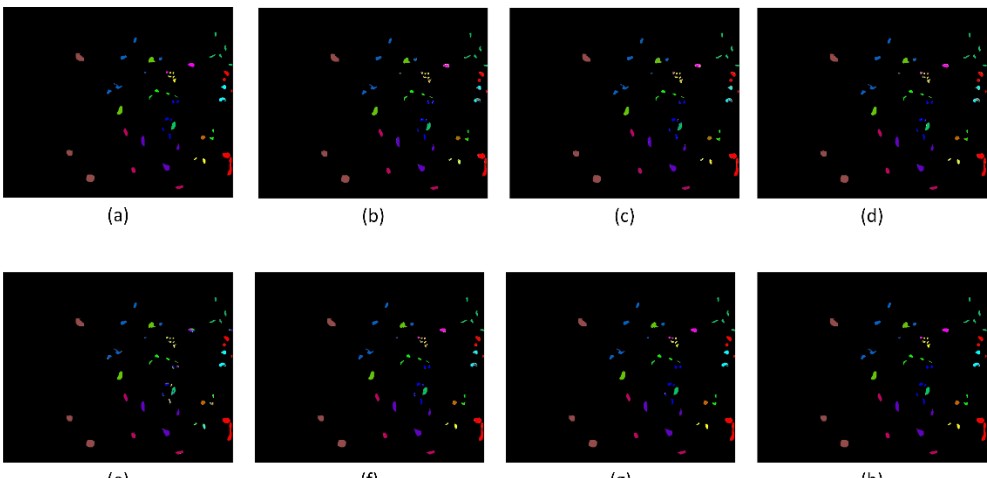

**Figure 12.** KSC dataset. (**a**) Ground truth; (**b**) SVM; (**c**) AENSVM; (**d**) TSSVM; (**e**) ResNet; (**f**) SSRN; (**g**) A2S2K-ResNet; and (**h**) OSANet-TSSVM.

Figure 12 shows that among the seven experimental sets, OSANet-SVM exhibits significantly fewer classification errors than other algorithms. Particularly in CP hammock, oak/broadleaf, and graminoid marsh categories, the clarity of the classification image boundaries and the notably reduced number of misclassified points are apparent, aligning closely with the ground truth maps. As demonstrated in Table 10, OSANet-SVM achieves the best classification results, with the three comprehensive evaluation metrics of AA, OA, and Kappa coefficients reaching 98.75%, 97.94%, and 98.63%, respectively. Furthermore, the results indicate that the KSC dataset comprises 13 categories, and OSANet-SVM achieves a classification accuracy of 100% in six of these categories. This demonstrates that OSANet-SVM effectively captures the feature relationships in the input images, optimises classifier performance, and enables the extraction of spatial global features from the imagery.

### 3.6. Ablation Experiment

Ablation experiments were conducted to validate the effectiveness of the OSAM, CAM, and TSSVM modules for hyperspectral land classification tasks; the results are presented in Table 11. The highest accuracy values are highlighted in bold. In the table, (1) represents the pyramid residual backbone network, and (2) and (3) indicate the addition

of OSAM and CAM modules to the backbone network, respectively, (4) and (5) represent the removal of the CAM and TSSVM modules, respectively, and (6) represents the proposed OSANet-TSSVM model in this study.

**Table 11.** Ablation experiment.

| | Model | (1) | (2) | (3) | (4) | (5) | (6) |
|---|---|---|---|---|---|---|---|
| | OSM | - | √ | - | √ | √ | √ |
| | CAM | - | - | √ | - | √ | √ |
| | TSSVM | - | - | - | √ | - | √ |
| Indian Pine | OA | 97.27 | 97.79 | 97.55 | 98.01 | 98.06 | **98.19** |
| | AA | 96.43 | 97.04 | 96.23 | 97.83 | 97.97 | **97.73** |
| | Kappa | 97.36 | 97.61 | 97.77 | 97.66 | 98.36 | **98.13** |
| PaviaU | OA | 97.31 | 97.40 | 97.35 | 97.81 | 98.39 | **98.64** |
| | AA | 96.29 | 96.69 | 96.83 | 97.08 | 97.50 | **98.34** |
| | Kappa | 97.21 | 97.89 | 96.34 | 97.85 | 98.48 | **98.17** |
| Salinas | OA | 97.64 | 97.78 | 97.70 | 97.97 | 98.18 | **98.32** |
| | AA | 98.65 | 98.75 | 98.04 | 99.05 | 99.01 | **99.29** |
| | Kappa | 97.12 | 97.94 | 97.17 | 97.29 | 97.41 | **98.11** |

From the experimental results, it is evident that all the proposed modules enhance the classification performance. Taking Indian Pine as an example, adding the OSAM module increased OA, AA, and Kappa by 0.52%, 0.61%, and 0.24%, respectively. Furthermore, the inclusion of both OSAM and TSSVM modules further improved classification performance. Incorporating the OSAM and CAM modules yielded evaluation indicators of 98.06%, 97.97%, and 98.36%, respectively.

These findings demonstrate that the attention mechanism in the OSAM and CAM modules successfully extracted more accurate spatial–spectral information. In addition, the TSSVM module optimised the discriminative results of feature extraction using the deep network, significantly enhancing the hyperspectral land classification performance of the model.

## 4. Conclusions

This paper presents a novel hyperspectral image classification algorithm called OSA-SVM, which combines deep learning and SVM. The algorithm enhances the deep feature extraction network and the SVM classification module to achieve more accurate classification outcomes. Based on the window sliding convolution operation, the convolutional neural network model considers only local information correlation within the image. However, deepening the network to obtain a larger receptive field introduces numerous redundant parameters, leading to an overfitting. In contrast, the visual transformer model, which utilises the self-attention mechanism, can capture global interdependence among pixel information. Nonetheless, the computational complexity of this method increases exponentially with image resolution, restricting its practical application. To address this problem, this study proposes a lightweight orthogonal self-attention mechanism that captures global features. This approach exclusively performs global self-attention calculations in two orthogonal spatial dimensions, thereby reducing the number of parameters while preserving the interaction of global contextual information. Additionally, a channel attention mechanism was introduced to improve the extraction of spectral feature information, further enhancing classification accuracy.

In the classification module, the traditional SVM often performs poorly in scenarios involving overlapping categories or non-linear separability. In addition, noise and outliers can lead to the overfitting of the model. To address these limitations, this paper presents an enhanced SVM model that incorporates a two-step training process. In the first stage, a discriminant function was obtained through SVM training. This function was then retained as a new feature for the second-level training, resulting in the final TSSVM model. By

jointly optimising the two classifiers and leveraging comprehensive feature information, the shortcomings of a single classifier are overcome, surpassing the performance of traditional SVM algorithms. Although the proposed method demonstrates excellent performance on multiple public datasets, it is essential to note that data compatibility between deep networks and traditional methods may affect the stability of the algorithm. As a result, future research will focus on further reducing the model parameter quantity and computational complexity, as well as improving the robustness of the hybrid model algorithm.

Our model is feasible for application in real-world scenarios. Although all experiments in this paper were conducted on small hardware devices and limited datasets, the deployment in practical scenarios can be achieved. The lightweight design proposed in this paper will significantly reduce inference time to apply the model to real-time processing scenarios. Of course, hyperspectral images collected in real-world scenarios contain more complex image degradation problems, such as blurring, occlusion, etc. To overcome this problem, targeted preprocessing should be performed on the images before the classification task, such as image denoising, image super-resolution, etc. Suppose the model is to be applied in particular environments, such as underwater, high temperature, etc. In that case, different hardware materials should be selected and continuously tested to ensure the stability of the model.

**Author Contributions:** Conceptualisation, L.W.; methodology, H.S.; formal analysis, Y.S.; writing, H.L. All authors have read and agreed to the published version of the manuscript.

**Funding:** This research was funded by the National Natural Science Foundation of China (grant number 62071084), and by the Leading Talents Project of the State Ethnic Affairs Commission.

**Data Availability Statement:** All datasets can be obtained at http://www.ehu.eus/ccwintco/index.php?title=Hyperspectral_Remote_Sensing_Scenes (accessed on 20 May 2011).

**Conflicts of Interest:** The authors declare no conflicts of interest.

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
