# Peer review of "Hyperspectral Image Classification with the Orthogonal Self-Attention ResNet and Two-Step Support Vector Machine"

_remotesensing, doi:10.3390/rs16061010_

Round 1

Reviewer 1 Report

Comments and Suggestions for Authors

Comments to remotesensing-2861267

General comment: 

This article presents a novel hyperspectral image classification framework, OSANet-TSSVM, addressing the limitations of existing methods in spatial global information interaction and feature extraction. Experimental results on Indian Pine, Salinas, and Pavia University datasets demonstrate the superiority of OSANet-TSSVM over seven existing methods in both subjective and objective evaluation metrics. Minor revisions are required before publication to address detailed issues outlined below.

Detail comments:

1.    The format of parentheses in Eq.2 is inconsistent with other formulas. Additionally, if LN is an operation, should “(∙)” be added? Please carefully check.

2.    Chapter 3 lacks a subtitle for the Salinas dataset; please add it.

3.    The experiment did not reflect the Kennedy Space Center datasets mentioned in the experimental analysis section. Please ensure completion of this experiment.

4.    Fig 1 does not specify the implementation details of Layer N. Are the parameter settings consistent between the three Layers besides the different downsampling? Please provide clarification.

5.    The ablation experiment states that the OSANet-TSSVM model achieved varying degrees of accuracy improvement on three datasets, but fails to mention the practical significance of hyperspectral image classification. This will make it difficult for readers to have a concrete understanding of the research value of the work. Please address this point.

Comments on the Quality of English Language

Moderate editing of English language required

Reviewer 2 Report

Comments and Suggestions for Authors

ThThere are fundamental problems in this manuscript, so I can't recommend to accept this manuscript in its' present form:

1. What are SSRN, A2S2K-ResNet, there are no references for these compared methods. It is very difficult for readers to understand without any references.

2. There are abundament of HSI classification methods recently. The compared baselines in the manuscript are too old, I couldn't see any advantages to compare with RF SVM.

Comments on the Quality of English Language

In my opinion, there are some fundamental problems of this manuscript, I couldn't recommend to accept this manuscript in its' present form.

Reviewer 3 Report

Comments and Suggestions for Authors

Dear Authors,

The paper presents a novel hyperspectral image classification algorithm, OSA-SVM, which integrates deep learning and SVM to enhance classification accuracy. The algorithm addresses the limitations of traditional methods by incorporating a lightweight orthogonal self-attention mechanism for global feature extraction and a two-step training process for improving SVM performance.

What I find stronger of the paper is that it includes its innovative approach to hyperspectral image classification, comprehensive evaluation on multiple public datasets, and incorporating advanced techniques such as global self-attention and TSSVM. These contributions show a deep understanding of the challenges in hyperspectral image processing and offer practical solutions to improve classification accuracy.

However, there are weaknesses in the paper, such as the limited experimental validation on publicly available datasets, the lack of a detailed comparative analysis with modern techniques, and the need for a more thorough exploration of scalability issues. Additionally, practical considerations for implementing the proposed algorithm in real-world scenarios could be better addressed.

Still, I suggest to improve parts to become an important contribution to the field of hyperspectral image processing. These parts are the following

1.       Expanded Experimental Validation: Expanding the experimental validation of the OSA-SVM algorithm involves conducting experiments on a wider range of hyperspectral image datasets that encompass varying resolutions, sensor types, and environmental conditions. This entails careful dataset choice to ensure diversity and representativeness, followed by rigorous preprocessing to standardize the data and enhance its quality. Clear experimental protocols should be defined, including partitioning datasets and documenting parameters. By applying the algorithm to each dataset and evaluating its performance using standard metrics, researchers can gain insights into its robustness and generalizability across different scenarios. Analyzing the results allows for a comprehensive understanding of the algorithm's strengths, weaknesses, and applicability. Through interpretation and discussion, researchers can elucidate the algorithm's performance trends and identify areas for improvement. This expanded validation not only enhances confidence in the algorithm's effectiveness but also guides future research directions to address any identified limitations and advance the field of hyperspectral image processing.

2.       Detailed Comparative Analysis: To accomplish a detailed comparative analysis with modern techniques in hyperspectral image processing, several steps can be taken. First, it's essential to identify a comprehensive set of existing methods that represent the current state-of-the-art in the field. This involves conducting a thorough literature review and selecting a diverse range of algorithms that address similar tasks or problems as the proposed OSA-SVM algorithm. Next, establish clear criteria for comparison, considering relevant parts such as accuracy, computational efficiency, robustness to noise, scalability, and ease of implementation. Then, apply each selected method, including the OSA-SVM algorithm, to the same set of datasets under standardized conditions to ensure fair comparison. Utilize performance metrics to evaluate and measure the performance of each algorithm objectively. Finally, analyse the results, comparing the strengths and weaknesses of the OSA-SVM algorithm with those of existing methods. Highlight any unique advantages or limitations of the proposed algorithm and provide insights into its potential practical implications and areas for further improvement. This detailed comparative analysis will provide valuable insights into the relative performance and applicability of the OSA-SVM algorithm compared to modern techniques, aiding researchers and practitioners in making informed decisions about its adoption and further development.

3.       Scalability Analysis: To conduct a scalability analysis of the algorithm, several steps should be followed. First, identify a variety of hyperspectral datasets of differing sizes, ranging from small-scale to large-scale, representing the range of data sizes typically found in real-world applications. Next, implement the OSA-SVM algorithm using different dataset sizes and assess its computational resource requirements, such as memory usage and processing time, for each dataset size. This evaluation should include both training and inference phases to comprehensively understand the algorithm's resource demands. Additionally, consider conducting experiments on different hardware configurations, such as CPUs, GPUs, or specialized accelerators, to evaluate how the algorithm's performance scales with different computational resources. Analyze the results to identify any scalability bottlenecks, such as memory constraints or computational inefficiencies, and propose potential optimizations or parallelization strategies to improve scalability. Finally, provide insights into the algorithm's feasibility for real-world applications with large-scale hyperspectral datasets based on the scalability analysis results, highlighting any limitations or areas for improvement to guide future research and development efforts.

4.       Discussion on Practical Implementation: Provide insights into practically implementing the proposed algorithm, including considerations for deployment in real-world scenarios, potential challenges, and strategies for overcoming them.

5.       Address Potential Limitations: Discuss potential limitations and constraints of the proposed algorithm, such as data compatibility issues between deep learning and traditional methods, and propose strategies for mitigating these limitations in future research.

The paper presents an important contribution to the field of hyperspectral image processing by proposing an innovative algorithm that improves classification accuracy. Addressing the identified weaknesses would further strengthen the paper and enhance its impact in the research community.

Round 2

Reviewer 2 Report

Comments and Suggestions for Authors

After carefully reviewing, I can notice that the authors have addressed all my questions, and I have no other extended questions.

Reviewer 3 Report

Comments and Suggestions for Authors

Dear authors,

Thank you for your detailed response to my feedback on the manuscript. I appreciate the efforts you have made in revising the manuscript to address the concerns raised.

I'm glad to see that you have tried to improve the paper, particularly in response to the suggestions regarding diversity, comparative analysis, scalability analysis, discussion on practical implementation, and addressing potential limitations. Incorporating the Kennedy Space Center dataset adds diversity to the experimental data and strengthens the generalizability of the findings. Your efforts to provide insights into practical implementation considerations and address potential limitations show a thorough understanding of the broader implications of the proposed algorithm.

Best regards,